# An intrinsic tumour eviction mechanism in *Drosophila* mediated by steroid hormone signalling

Yanrui Jiang [1], Makiko Seimiya[1], Tommy Beat Schlumpf[1] & Renato Paro [1,2]

Polycomb group proteins are epigenetic regulators maintaining transcriptional memory during cellular proliferation. In *Drosophila* larvae, malfunction of Polyhomeotic (Ph), a member of the PRC1 silencing complex, results in neoplastic growth. Here, we report an intrinsic tumour suppression mechanism mediated by the steroid hormone ecdysone during metamorphosis. Ecdysone alters neoplastic growth into a nontumorigenic state of the mutant *ph* cells which then become eliminated during adult stage. We demonstrate that ecdysone exerts this function by inducing a heterochronic network encompassing the activation of the microRNA *lethal-7*, which suppresses its target gene *chronologically inappropriate morphogenesis*. This pathway can also promote remission of brain tumours formed in *brain tumour* mutants, revealing a restraining of neoplastic growth in different tumour types. Given the conserved role of *let-7*, the identification and molecular characterization of this innate tumour eviction mechanism in flies might provide important clues towards the exploitation of related pathways for human tumour therapy.

[1] D-BSSE, ETH Zürich, Mattenstrasse 26, 4058 Basel, Switzerland. [2] Faculty of Sciences, University of Basel, Klingelbergstrasse 50, 4056 Basel, Switzerland. Correspondence and requests for materials should be addressed to R.P. (email: renato.paro@bsse.ethz.ch)

Polymcomb group (PcG) proteins are evolutionarily conserved chromatin regulators modulating histone modifications and suppressing target gene expression, required for the maintenance of cellular memory[1,2]. PcG proteins can bind to particular genomic regions, where they mediate specific histone modifications and chromatin compaction, therefore suppressing the expression of the target genes in these loci. Dysregulation of PcG genes is associated with various human cancers, but the mechanisms are incompletely understood[3].

PcG proteins form two major Polycomb repressive complexes, PRC1 and PRC2, to silence the expression of target genes. Previous studies have shown that the PRC1 components can act as tumour suppressors in *Drosophila*[4,5]. In the developing eye-antennal imaginal discs, for instance, cells homozygous mutant for *ph* overgrow and give rise to neoplastic tumours[4,5]. These tumours can be transplanted and continue to grow in wild-type adult flies[5].

Here, we carry out studies to investigate the mechanisms underlying tumour formation and growth in *ph* mutants. Unexpectedly, we observe that the tumorigenic *ph* mutant cells are transformed into nontumorigenic cells after metamorphosis, and eventually evicted in adult flies. We show that ecdysone signalling is responsible for the transformation of tumorigenicity. By performing transcriptome analyses we identify miRNA *let-7* as a key target of the ecdysone response in this process. We further demonstrate that mis-expression of *chronologically inappropriate morphogenesis* (*chinmo*), a direct target of both Ph and *let-7*, is required for tumour growth. Furthermore, we show that the *let-7* cascade could also suppress the overgrowth of brain tumours in *brain tumour* (*brat*) mutant flies. Our analyses reveal an intrinsic mechanism that is able to reprogram tumorigenic cells and suppress their malignant growth in adult *Drosophila*.

## Results

### Conversion of tumorigenic $ph^{505}$ cells during metamorphosis.
The *Drosophila* genome encodes two *ph* genes, *ph proximal* (*ph-p*) and *ph distal* (*ph-d*)[6]. $ph^{505}$ is a loss of function allele of both genes[7]. Homozygous $ph^{505}$ clones, generated genetically by MARCM (mosaic analysis with a repressible cell marker)[8] and marked by GFP, overgrow and give rise to large tumours in the larval eye-antennal discs at the wandering third instar (Fig. 1a). The morphology of these clones is in sharp contrast to wild-type GFP-expressing clones (Fig. 1b). After transplanting $ph^{505}$ eye disc tumours into wild-type adult hosts (Fig. 1c, arrow), $ph^{505}$ cells continued to proliferate, resulting in the formation of neoplastic tumours (Fig. 1c, d). This indicates that larval $ph^{505}$ cells are tumorigenic and is also consistent with previously reported results[4,5]. These tumours can recapitulate proliferation after serial retransplantation into new hosts, but they did not give rise to metastatic tumours in other parts of the body (Fig. 1d). In newly eclosed adult flies, GFP-marked $ph^{505}$ cells can be observed all over the body, including the head, legs, thorax, and abdomen (Fig. 1e). However, this was caused by the expression of the *ey-flp* in all leg discs and the genital disc, producing GFP-marked clones in these tissues as well (see Methods).

In the head, these marked $ph^{505}$ cells formed grape-like, single-layered epithelial structures (Fig. 1f; Supplementary Fig. 1a). Surprisingly and in contrast to transplanted tumour tissue, the $ph^{505}$ tumour cells disappeared gradually during fly adulthood (Fig. 1g; Supplementary Fig. 1b). Immunostainings showed that the single layer of $ph^{505}$ cells in these spherical structures did not proliferate and did not differentiate into neurons (Fig. 1f). Moreover, after transplantation of these $ph^{505}$ structures into wild-type hosts (Fig. 1h, arrow), these cells did not grow and also disappeared within a few days (Fig. 1i). A subset of the $ph^{505}$ cells

found in adult flies expressed *Drosophila* cleaved death caspase-1 (cDCP-1) (Fig. 1j), an apoptosis cell marker[9]. In addition, the autophagy marker *UAS-mCherry:Atg8* (Ch:Atg8)[10] was also expressed in a number of cells (Fig. 1k). These results suggest that there is a conversion of tumorigenic larval $ph^{505}$ cells into nontumorigenic adult $ph^{505}$ cells at metamorphosis (henceforth named metamorphed cells), which are then eliminated by either apoptotic and/or autophagic cell death in the adult.

### Ecdysone controls the transformation of $ph^{505}$ tumour cells.
20-hydroxyecdysone (ecdysone) is the key molting steroid hormone controlling metamorphosis of flies[11]. Ecdysone is produced as a series of brief low-level pulses during embryonic and early larval stages. Near the end of third larval instar, a mid-level pulse of ecdysone triggers pupariation[11,12]. The expression of ecdysone increases and reaches the peak level around 48 h after pupa formation. Afterward, ecdysone expression gradually decreases to a low basal level, which is then maintained during adulthood[11,12]. To assess the role of ecdysone in altering the oncogenic potential of $ph^{505}$ cells, we ectopically expressed a dominant-negative form of the ecdysone receptor (EcR)[13] in $ph^{505}$ mutant cells ($ph^{505}$; *UAS-EcR^{DN}*). In contrast to metamorphed $ph^{505}$ cells (Fig. 1e, g), $ph^{505}$; *UAS-EcR^{DN}* cells continued to grow in the adults and resulted in the accumulation of large tumours throughout the body (Fig. 2a). In the head, the tumour mass (Fig. 2b, arrow) could reach a similar size as the adult brain (Fig. 2c, arrowhead). As a result, these flies showed a significantly reduced lifespan (Fig. 2d) and $ph^{505}$; *UAS-EcR^{DN}* cells were able to give rise to neoplastic tumours after transplantation (Fig. 2e). A knockdown of EcR co-receptor Ultraspiracle (Usp) using transgenic RNAi ($ph^{505}$; *UAS-RNAi-usp*) showed similar results substantiating ecdysone involvement (Supplementary Fig. 2). Hence, the manifestation of the ecdysone pulse at metamorphosis appears directly responsible for suppressing the tumorigenic character of larval $ph^{505}$ cells. Conversely, a cell-autonomous block of ecdysone signalling retains the tumorigenicity of $ph^{505}$ cells in the intact adult flies.

To better understand how ecdysone exerts its tumour-suppressor function in $ph^{505}$ cells, we performed transcriptome analyses by mRNA sequencing. RNA from metamorphed $ph^{505}$ cells in adults, 4, 8, 14 weeks old tumours in adult hosts after weekly re-transplantations, and wild-type transplanted discs as control were extracted and sequenced. In total, we identified 2015 significantly differentially expressed genes. Applying k-means clustering on the gene expression profiles across wild-type, tumour and metamorphed samples, groups of genes with similar profiles can be found (Fig. 2f; Supplementary Fig. 3). Gene ontology (GO) term analysis of the resulting five groups revealed that transcription regulators and genes required for metabolic processes were upregulated in the tumours, but genes involved in neuronal differentiation were downregulated (Fig. 2f; Supplementary Fig. 3). Cell-cycle-related genes were downregulated in metamorphed $ph^{505}$ cells in addition to the differentiation-related genes (Fig. 2f; Supplementary Fig. 3), when compared to a wild-type transplanted disc. Lower levels of neuronal marker Elav can corroborate this (Fig. 1f). We next focused on genes that are known to respond to ecdysone (Supplementary Fig. 4). Expression levels of a group of genes was upregulated in the metamorphed $ph^{505}$ cells but decreased in the transplanted tumours (Fig. 2g). Among these is *let-7-C*, which is a polycistronic locus encoding three miRNAs including *let-7*[14]. As the mammalian homologues of *let-7* are underexpressed in various cancer types[15], we decided to further investigate the potential role of *let-7* in altering $ph^{505}$ tumorigenicity.

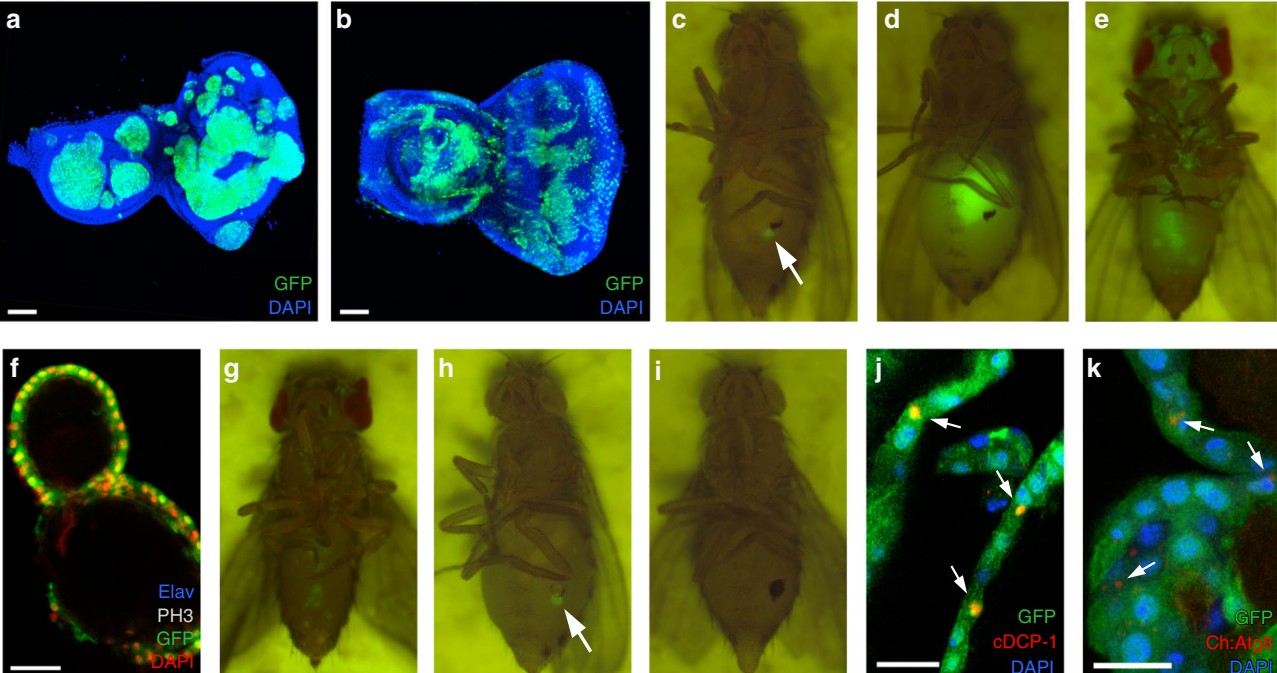

**Fig. 1** Conversion of tumorigenic $ph^{505}$ cells into nontumorigenic metamorphed cells. **a**, **b** Confocal images of the eye-antennal discs at wandering third instar containing overgrown $ph^{505}$ tumour (**a**) or wild-type clones (**b**). Scale bars are 50 μm. **c**, **d** Transplantation of a small piece of the eye disc containing GFP-labelled $ph^{505}$ cells (arrow) into a wild-type adult host. Pictures of the same host were taken at 1 day (**c**) or at 2 weeks (**d**) after transplantation, showing tumour formation in the abdomen (**d**). **e** GFP-labelled $ph^{505}$ cells are present throughout the body in the adults at 1 day after eclosion. **f** Confocal image showing the $ph^{505}$ cells form a single layer of cells in a grape-like structure. The cells do not proliferate (negative for PH3) and do not differentiate into neurons (negative for Elav). Scale bar is 20 μm. **g** The GFP-positive cells disappeared after 4 days in the same fly as **e**. **h**, **i** Transplantation of metamorphed $ph^{505}$ cells into a new wild-type adult host (arrow). Pictures of the same host were taken immediately after transplantation (**h**) or at 1 week after transplantation (**i**), showing the transplanted cells do not grow but disappear. **j**, **k** Confocal images of metamorphed $ph^{505}$ cells in the grape-like structures, showing a subset of cells expressing the apoptosis cell marker cDCP-1 (**j**, arrows) or the autophagy cell marker Ch:Atg8 (**k**, arrows). Scale bars are 20 μm. Genotypes: **a**, **e**, **f**, **g**, **j** $ph^{505}$ FRT19A/tub-Gal80 FRT19A; +/ey-flp act>STOP>Gal4 UAS-GFP; **b** y w FRT19A/tub-Gal80 FRT19A; +/ey-flp act>STOP>Gal4 UAS-GFP; **k** $ph^{505}$ FRT19A/tub-Gal80 FRT19A; UAS-mCherry:Atg8a/ey-flp act>STOP>Gal4 UAS-GFP

**Ecdysone-induced *let-7* suppresses the growth of $ph^{505}$ tumour.** In *Drosophila*, *let-7* expression is induced by ecdysone at the start of pupariation, and the expression of *let-7* correlates with the dynamic changes of the ecdysone level during metamorphosis[16]. A first step was to confirm mature *let-7* levels in the various samples using the Taqman quantitative PCR assay. As previously reported[16], the expression of mature *let-7* was low at the wandering third larval instar but increased significantly to a high level at 48 h after pupa formation (Fig. 3a). Indeed, *let-7* expression was also elevated in the metamorphed $ph^{505}$ cells, but not in the $ph^{505}$ transplanted tumours (Fig. 3a). Moreover, *let-7* level in $ph^{505}$; *UAS-EcR$^{DN}$* cells collected from adults remained low (Fig. 3a), indicating *let-7* was not induced when ecdysone signalling was blocked.

To further characterize the role of *let-7*, we reduced the activity of endogenous *let-7* in $ph^{505}$ cells using a transgenic miRNA sponge[17] ($ph^{505}$; *UAS-let-7-SP*) (Fig. 3a). We observed that $ph^{505}$; *UAS-let-7-SP* cells maintained the tumorigenic growth in the adults (Fig. 3b). After dissection and transplantation, $ph^{505}$; *UAS-let-7-SP* cells collected from adults retained tumour formation in the hosts (Fig. 3c), indicating that *let-7* is necessary to mediate the suppression of tumorigenic growth of $ph^{505}$ cells. Next, we overexpressed *let-7* in $ph^{505}$ cells. Overexpression of either the entire *let-7-C* ($ph^{505}$; *UAS-let-7-C*) or *let-7* alone ($ph^{505}$; *UAS-let-7*) could strongly suppress tumour growth in the eye-antennal discs (Supplementary Fig. 5a, b). The average tumour volume was reduced to 21% in $ph^{505}$; *UAS-let-7* larvae (Fig. 3d). The effect of *let-7* overexpression appeared to be independent of apoptosis, as

overexpression of *let-7* did not lead to elevated cell death in the eye-antennal discs (Supplementary Fig. 5c, d). Moreover, $ph^{505}$; *UAS-let-7* cells did not give rise to tumours after transplantation into adult hosts ($n > 50$).

*let-7-C* also encodes another two miRNAs, *miR-100* and *miR-125*. We next tested if these two miRNAs might also play a role in transforming the tumorigenic $ph^{505}$ cells. First, unlike *let-7-SP*, $ph^{505}$; *UAS-miR-100-SP* cells stopped growing and disappeared in the adult flies (Supplementary Fig. 6a). Immunostaining showed that the apoptosis cell marker cDCP-1 was expressed in the $ph^{505}$; *UAS-miR-100-SP* cells (Supplementary Fig. 6b), suggesting that the $ph^{505}$; *UAS-miR-100-SP* cells can still be transformed into nontumorigenic cells and undergo apoptosis. For *miR-125* we observed that the $ph^{505}$; *UAS-miR-125-SP* cells stopped growing in the adult flies, but were still visible in these flies 2 weeks after eclosion (Supplementary Fig. 6c), unlike the metamorphed $ph^{505}$ cells that eventually are eliminated. Immunostaining showed that $ph^{505}$; *UAS-miR-125-SP* cells did not express the apoptosis cell marker cDCP-1 (Supplementary Fig. 6d), nor did they express the mitosis marker PH3 (Supplementary Fig. 6e). Furthermore, when we transplanted adult $ph^{505}$; *UAS-miR-125-SP* cells into host flies (Supplementary Fig. 6f), they did not give rise to neoplastic tumours, indicating the $ph^{505}$; *UAS-miR-125-SP* cells are not tumorigenic anymore. However, even at 4 weeks after transplantation, the $ph^{505}$; *UAS-miR-125-SP* cells were still present (Supplementary Fig. 6g). These results show that $ph^{505}$; *UAS-miR-125-SP* cells neither proliferate nor undergo apoptotic cell death.

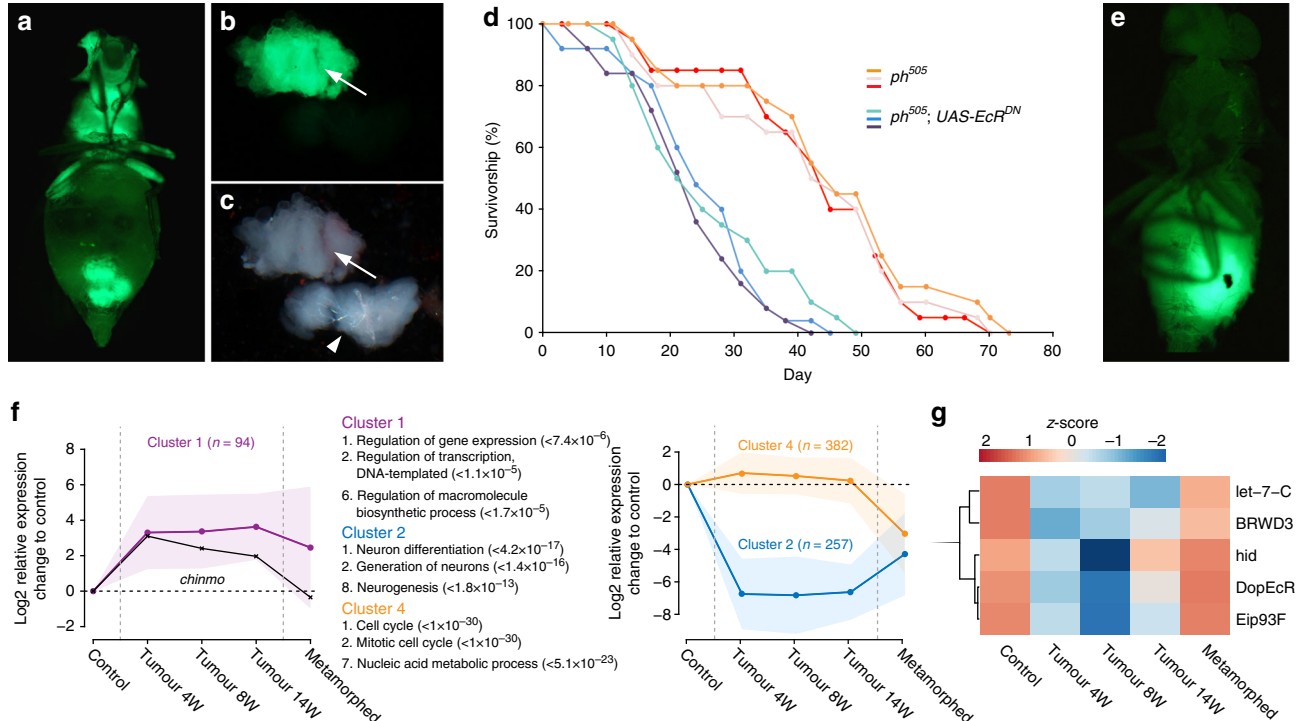

**Fig. 2** Ecdysone controls the conversion into metamorphed $ph^{505}$ cells. **a** An adult fly of $ph^{505}$; $UAS\text{-}EcR^{DN}$ 3 weeks after eclosion, showing the formation of large tumours throughout the body. **b**, **c** Dissected tumour mass (arrow) from the head of an adult fly as in **a**; note the size of the tumour is similar to that of the adult brain (arrowhead in **c**). **d** Survival of $ph^{505}$ and $ph^{505}$; $UAS\text{-}EcR^{DN}$ adult flies, showing a strongly reduced lifespan of $ph^{505}$; $UAS\text{-}EcR^{DN}$ adults. For each genotype, three independent experiments were carried out and indicated by different colours (see Methods for details). **e** Formation of tumours in a wild-type host after transplantation of $ph^{505}$; $UAS\text{-}EcR^{DN}$ cells dissected from the head. **f** Clustering of significantly differentially expressed genes and their associated GO-terms. Cluster 1 consists of transcription regulators that are upregulated in tumours, some of which return to pretumour levels in metamorphed $ph^{505}$ cells. Cluster 2 describes genes required for neuron differentiation which are downregulated in both tumours and metamorphed $ph^{505}$ cells, but cluster 4 consists of genes involved in cell cycle that are only downregulated in the metamorphed $ph^{505}$ cells. All $n$ genes of the respective cluster can be found within the shaded area, whereas the lines denote the mean. The complete clustering analysis result is shown in Supplementary Figure 3. **g** Heat map of five ecdysone responsive genes whose expression was decreased in tumours, but upregulated in the metamorphed $ph^{505}$ cells. The complete heat map of the expression of all genes known to respond to ecdysone is shown in Supplementary Fig. 4. Genotypes: **a**, **b**, **c** $ph^{505}$ $FRT19A$/$tub\text{-}Gal80$ $FRT19A$; $UAS\text{-}EcR^{DN}$/$ey\text{-}flp$ $act$>$STOP$>$Gal4$ $UAS\text{-}GFP$; **d** $ph^{505}$ $FRT19A$/$tub\text{-}Gal80$ $FRT19A$; $+$/$ey\text{-}flp$ $act$>$STOP$>$Gal4$ $UAS\text{-}GFP$ and $ph^{505}$ $FRT19A$/$tub\text{-}Gal80$ $FRT19A$; $UAS\text{-}EcR^{DN}$/$ey\text{-}flp$ $act$>$STOP$>$Gal4$ $UAS\text{-}GFP$

Because miRNA sponges inhibit miRNA/mRNA interaction by sequestration, the function of targeted miRNAs might not be disrupted completely. Therefore, we tested if the miRNA sponges used in our experiments were functional. In flies, the zinc finger transcription factor *chinmo* is a known target of *let-7* and *miR-125*[18]. Using quantitative PCR, we determined the expression of *chinmo*, and some computationally predicted targets of *miR-125* and *miR-100*, as there are no experimentally validated miRNA/mRNA interactions for these two miRNAs. The expression of *chinmo* indeed increased in $ph^{505}$; $UAS\text{-}let\text{-}7\text{-}SP$ cells (Supplementary Fig. 6h), but did not in $ph^{505}$; $UAS\text{-}miR\text{-}125\text{-}SP$ cells (Supplementary Fig. 6i). However, the expression of the two computationally predicted targets, *Zasp52* and *RecQ4*, increased in $ph^{505}$; $UAS\text{-}miR\text{-}125\text{-}SP$ cells (Supplementary Fig. 6i). On the other hand, the expression of predicted targets of *miR-100* did not increase in $ph^{505}$; $UAS\text{-}miR\text{-}100\text{-}SP$ cells. These results indicate that the *let-7* sponge and *miR-125* sponge appear to be functional in $ph^{505}$ cells, whereas the *miR-100* sponge may not.

To further evaluate the role of *miR-125* and *miR-100*, we carried out overexpression experiments. First, overexpression of *miR-125* ($ph^{505}$; $UAS\text{-}miR\text{-}125$) could partially reduce the tumour volume in the eye-antennal discs (Supplementary Fig. 5a). However, when we transplanted the $ph^{505}$; $UAS\text{-}miR\text{-}125$ cells into wild-type hosts, we could still observe the formation of tumours in 52% of the hosts ($n = 31$), indicating $ph^{505}$; $UAS\text{-}miR\text{-}$

*125* cells are still tumorigenic. Similarly, overexpression of *miR-100* ($ph^{505}$; $UAS\text{-}miR\text{-}100$) only slightly reduced the tumour volume in the eye-antennal discs (Supplementary Fig. 5a) and did not alter the tumorigenicity.

Taken together, these analyses show that *let-7* is the key regulator in the *let-7-C*. The miRNA appears indispensable and needed for suppressing the tumorigenic character of $ph^{505}$ cells. While our tests do not identify a significant involvement of *miR-100*, *miR-125* does appear to have a certain role in the elimination of the metamorphed cells. Previous work has revealed that there are cross-regulatory relationships among the three miRNAs[19], which may explain the complicated phenotype of the $ph^{505}$; $UAS\text{-}miR\text{-}125\text{-}SP$ cells described above.

### Ph and *let-7* target *chinmo* is required for $ph^{505}$ tumour growth. To further elucidate the mechanism of how *let-7* controls the tumorigenesis of $ph^{505}$ cells, we searched for genes that are bound by Ph in their promoter region[20] and are *let-7* targets. Among others, we identified the gene *chinmo*[21] (Fig. 3e). The expression of *chinmo* was significantly increased in the $ph^{505}$ tumours but decreased in the metamorphed $ph^{505}$ cells (Fig. 2f, black curve). Immunostaining showed that Chinmo was ubiquitously expressed in the transplanted tumours (Fig. 3f) but was not detectable in metamorphed $ph^{505}$ cells (Fig. 3g). In the eye discs,

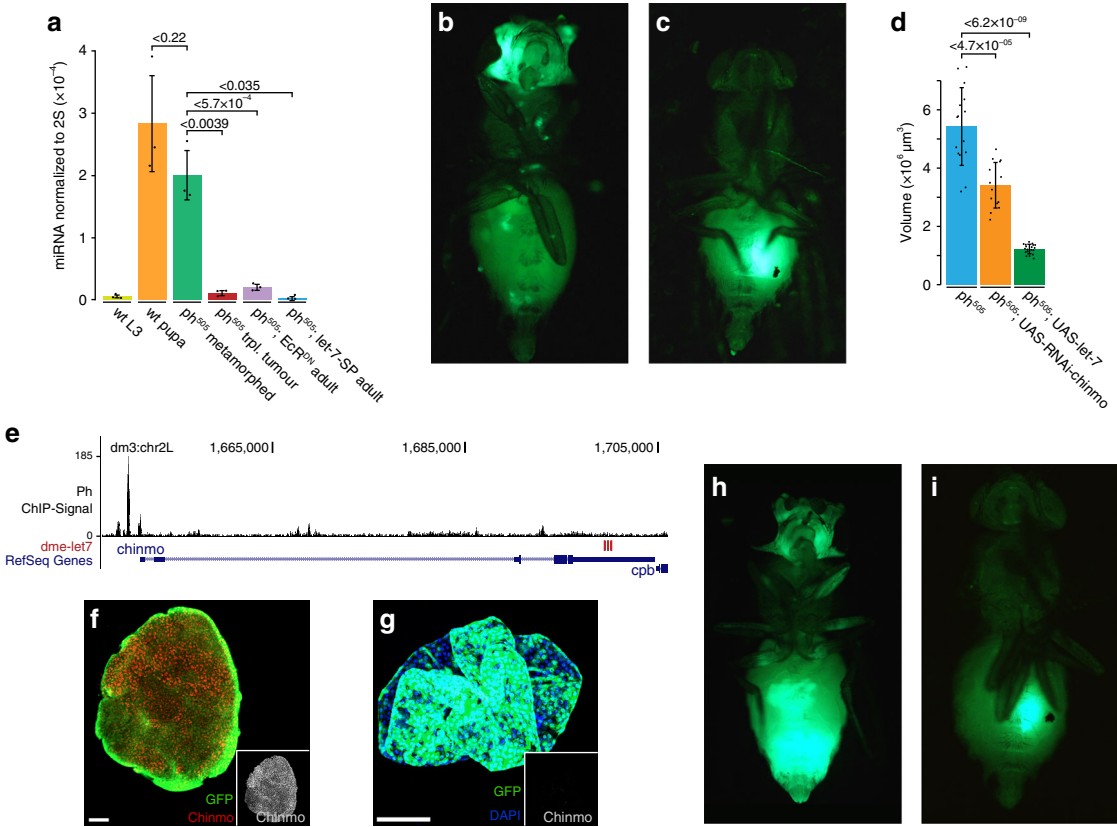

**Fig. 3** Ecdysone-induced *let-7* and its target Chinmo regulate the growth of *ph505* tumours. **a** Quantitative PCR analysis of the expression level of mature *let-7* in different samples. Mean of three of total three biological replicates with error bars describing the standard deviation. Statistical significance is determined by the *t*-test. **b** An adult fly of *ph505; UAS-let-7-SP* at 2 weeks after eclosion, showing the formation of tumours throughout the body. **c** Formation of tumours in a wild-type host after transplantation of *ph505; UAS-let-7-SP* cells dissected from the adult head. **d** Mean total volume of tumour clones in the eye-antennal discs of the different genotypes. *n* = 15 (*ph505*), 14 (*ph505; UAS-RNAi-chinmo*), and 19 (*ph505; UAS-let-7*), with error bars denoting the standard deviation. Significance was assessed using Welch's *t*-test. **e** Genome browser view of the *chinmo* locus, showing Ph ChIP signals at the transcription start site and several *let-7* binding sites (red bars) at the 3′UTR. **f, g** Confocal images of immunostaining with Chinmo antibodies. Chinmo is expressed in the transplanted tumour cells (**f**) but not in the metamorphed *ph505* cells (**g**). Scale bars are 50 μm. **h** An adult fly of *ph505; UAS-chinmo* at 2 weeks after eclosion, showing the formation of tumours all over the body. **i** Formation of tumours in a wild-type host after transplantation of *ph505; UAS-chinmo* cells dissected from the adult head. Genotypes: **a**, wt: *FM7 act-GFP/tub-Gal80 FRT19A; +/ey-flp act>STOP>Gal4 UAS-GFP*; *ph505*: *ph505 FRT19A/tub-Gal80 FRT19A; +/ey-flp act>STOP>Gal4 UAS-GFP*; *ph505; EcR^DN*: *ph505 FRT19A/tub-Gal80 FRT19A; UAS-EcR^DN/ey-flp act>STOP>Gal4 UAS-GFP*; *ph505; let-7-SP*: *ph505 FRT19A/tub-Gal80 FRT19A; UAS-let-7-SP/ey-flp act>STOP>Gal4 UAS-GFP; UAS-let-7-SP/+*; **b** *ph505 FRT19A/tub-Gal80 FRT19A; UAS-let-7-SP/ey-flp act>STOP>Gal4 UAS-GFP; UAS-let-7-SP/+*; **d** *ph505*: *ph505 FRT19A/tub-Gal80 FRT19A; +/ey-flp act>STOP>Gal4 UAS-GFP*; *ph505; UAS-RNAi-chinmo*: *ph505 FRT19A/tub-Gal80 FRT19A; +/ey-flp act>STOP>Gal4 UAS-GFP; UAS-RNAi-chinmo/+*; *ph505; UAS-let-7*: *ph505 FRT19A/tub-Gal80 FRT19A; +/ey-flp act>STOP>Gal4 UAS-GFP; UAS-let-7/+*; **f, g** *ph505 FRT19A/tub-Gal80 FRT19A; +/ey-flp act>STOP>Gal4 UAS-GFP*; **h** *ph505 FRT19A/tub-Gal80 FRT19A; +/ey-flp act>STOP>Gal4 UAS-GFP; UAS-chinmo/+*

Chinmo was expressed in the *ph505* mutant clones, as well as the neighbouring heterozygous cells (Supplementary Fig. 7a). However, at 48 h after pupa formation, when ecdysone level was at the peak and *let-7* was highly expressed (Fig. 3a), the expression of Chinmo became undetectable (Supplementary Fig. 7b). Another known *let-7* target is the *abrupt* gene[22], which has been shown to promote the development of some types of tumours in the eye-antennal discs[23]. However, by immunostaining we did not detect the expression of Abrupt in the transplanted *ph505* tumours, in the eye-antennal discs, as well as in the metamorphed *ph505* cells. This indicates that *abrupt* does not play a significant role in regulating the tumorigenesis of *ph505* cells.

To test the function of Chinmo, we overexpressed the protein in the tumour cells (*ph505; UAS-chinmo*) and observed that these cells continued to grow in the adult flies (Fig. 3h) and gave rise to tumours after transplantation (Fig. 3i). Furthermore, when *chinmo* was knocked down (*ph505; UAS-RNAi-chinmo*), the average volume of the *ph505; UAS-RNAi-chinmo* clones in the eye-antennal discs was reduced to 60% (Fig. 3d, Supplementary Fig. 5e). After transplantation of *ph505; UAS-RNAi-chinmo* eye discs, tumours formed in only 16% of the hosts (*n* = 67), significantly less frequent than after transplantation of larval *ph505* eye discs (79%, *n* = 74). These results show that *chinmo* is an important downstream effector of *let-7* in regulating the proliferation of *ph505*-driven tumours.

**Ecdysone-induced *let-7* suppresses the growth of brain tumours.** Because *let-7* expression is induced by ecdysone in a variety of tissues during metamorphosis[16], we next tested if *let-7*-dependent tumour suppression also acts in a different tissue. In *Drosophila*, *brat* gene mutations lead to the formation of malignant brain tumours[24]. To generate such tumours, we used *wor-Gal4 ase-Gal80* to knockdown *brat* (*UAS-RNAi-brat*) in the type II neuroblasts[25]. Compared to the wild-type adult brain (Fig. 4a), *brat* RNAi resulted in the growth of large brain tumours in all

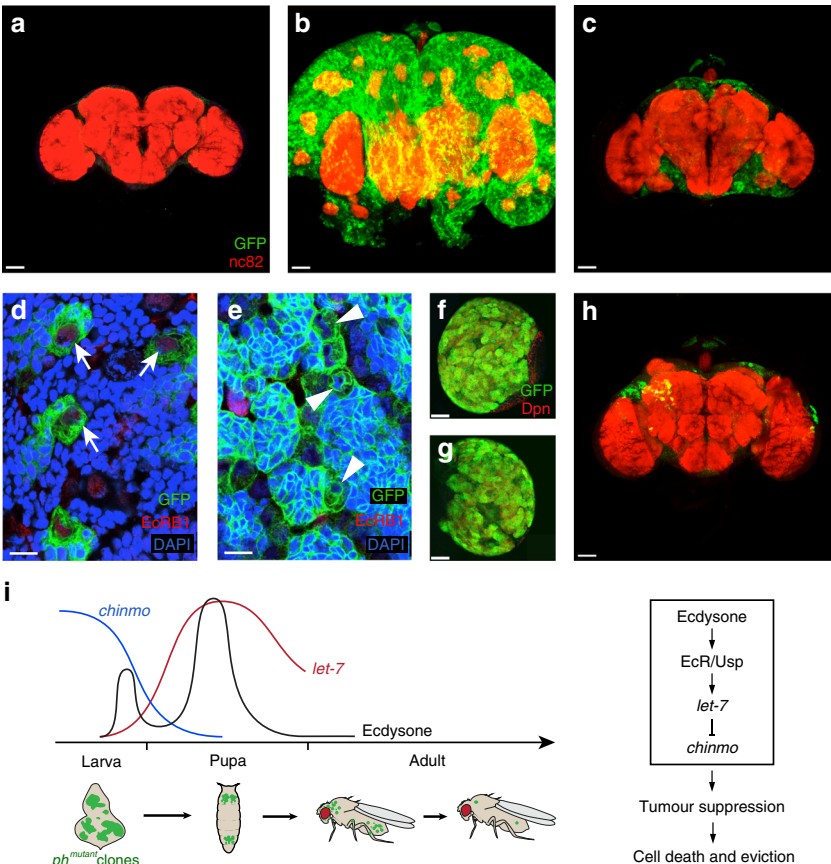

**Fig. 4** Eviction of *brat* tumours in adult brains by expression of *let-7* or ecdysone receptors. **a−c** Confocal images of adult brains of wild-type (**a**), *UAS-RNAi-brat* (**b**), and *UAS-RNAi-brat; UAS-let-7* (**c**). The brains are stained with the nc82 antibody (red) revealing brain structures, and GFP (green) that is expressed in the brain tumour cells. Note that the number of tumour cells is clearly reduced and the growth of tumour is greatly suppressed in *UAS-RNAi-brat; UAS-let-7* brains (**c**). Scale bars are 50 μm. **d, e** Confocal images of third instar larval brains of wild-type (**d**) and *UAS-RNAi-brat* (**e**). EcRB1 is expressed weakly in the type II neuroblasts in wild-type brains (**d**, arrows), but not in the tumour cells in *UAS-RNAi-brat* brains (**e**, arrowheads). Cells strongly express EcRB1 in (**d**) and (**e**) are glia cells that are not labelled by GFP. Scale bars are 10 μm. **f** Confocal image of a larval brain hemisphere of *UAS-RNAi-brat*, showing numerous tumour cells (green) that express Deadpan (Dpn) (red). Scale bar is 50 μm. **g** Confocal image of a larval brain hemisphere of *UAS-RNAi-brat; UAS-EcRB1*; note that a large tumour still forms. Scale bar is 50 μm. **h** Confocal image of an adult brain of *UAS-RNAi-brat; UAS-EcRB1*; note that the number of green brain tumour cells is significantly reduced. Scale bar is 50 μm. **i** Model for the intrinsic tumour eviction mechanism mediated by ecdysone-induced miRNA *let-7* in *ph505* tumours. In the drawing, ecdysone refers to 20-HE titres during prepupal and pupal stages[11,12]; *chinmo* and *let-7* refer to the mRNA expression in the eye-antenna discs. Genotypes: **a, d** *w UAS-dicer2; wor-Gal4 ase-Gal80; UAS-mCD8::GFP*; **b, e, f** *w UAS-dicer2; wor-Gal4 ase-Gal80/UAS-RNAi-brat; UAS-mCD8::GFP/+*; **c** *w UAS-dicer2; wor-Gal4 ase-Gal80/UAS-RNAi-brat; UAS-mCD8::GFP/UAS-let-7*; **g, h** *w UAS-dicer2; wor-Gal4 ase-Gal80/UAS-RNAi-brat; UAS-mCD8::GFP/UAS-EcRB1*

larvae and hatching adult flies (Fig. 4b). However, when *let-7* was overexpressed in *brat*-negative cells (*UAS-RNAi-brat; UAS-let-7*), a strong suppression of tumour growth in all adult brains was observed (Fig. 4c). Chinmo has recently been shown to play a key role in sustaining the tumour growth in *brat* mutants, and the overgrowth is markedly inhibited in *brat; chinmo* double mutant cells[26]. We suspected that *let-7* inhibits the *brat* tumour growth also by suppressing *chinmo*. However, since the *brat* mutant cells continue to grow in adult flies, we wondered why these tumours cannot be suppressed by the ecdysone pulse during metamorphosis as seen in *ph505* tumours. A recent study showed that the expression of the neuronal-specific ecdysone receptor isoform EcRB1 is temporally regulated in the neuroblasts; it is expressed from the mid third instar, when Chinmo is no longer expressed[27]. We therefore speculated that in *brat* tumours, ecdysone receptors are not expressed because of the sustained expression of Chinmo. Indeed, immunostaining with EcRB1-specific antibodies showed that the receptor was only weakly expressed in wild-type type II neuroblasts (Fig. 4d, arrows) and undetectable in *brat* tumours (Fig. 4e). Consequently, we tested whether ectopic expression of

the EcRB1 in the tumour cells would restore the eviction mechanism. In both cases, larvae of the genotype *brat*-RNAi as well as larvae coexpressing EcRB1 (*UAS-RNAi-brat; UAS-EcRB1*) show large brain tumours at the third larval instar (Fig. 4f, g). However, in the case of EcRB1 coexpression, the size of the brain tumours was significantly reduced in all adult brains (Fig. 4h), demonstrating that also *brat* tumour cells can be metamorphed. Indeed, the pupal ecdysone/*let-7* pulse induces a substantial remission of the tumorous tissue restoring almost wild-type brain morphology (Fig. 4h).

## Discussion

Deregulation of PcG gene expression has been associated with various types of human cancer[3,28]. For example, loss of expression of the human *ph* homologue has been linked to the formation of osteosarcomas[29,30]. As the molecular mechanisms of PcG proteins in human cancers are largely unknown, understanding the tumour-suppressor function of PcG genes in *Drosophila* therefore could provide insights in human cancer biology. During

the past decades imaginal discs have been used as a powerful paradigm to investigate mechanisms underlying the formation and progression of several types of tumour, including Ras-, PcG-, or Hippo pathway-induced tumours[31]. In addition, it is worth noting that the *let-7* consensus sequence is identical from *Caenorhabditis elegans* to humans, suggesting that *let-7* may control functionally conserved targets in regulating proliferation and differentiation during development[15,32,33]. In various types of human cancer, downregulation of one or more *let-7* members has been observed[33–37]. Moreover, induced expression of *let-7* in cancer cell lines can suppress cell proliferation and tumour growth[33,38]. In the human genome, the *let-7* family consists of more than ten members. However, the transcriptional regulation, spatial and temporal expression, and their tissue-specific and/or redundant functions of the *let-7* family in human are far more complicated and still remain elusive[15,39].

Here, we identified an intrinsic mechanism reprogramming tumorigenic to nontumorigenic cells of at least two different tumour types, by marking the cells for destruction in adult *Drosophila* (Fig. 4i). We found that the steroid hormone-induced miRNA *let-7* is a key mediator of this mechanism. Interestingly, *let-7* and its target genes, including *chinmo* have been shown to act as heterochronic genes that regulate developmental transitions[40]. Other findings from our lab indicate that $ph^{505}$ tumour cells are reprogrammed from the original larval imaginal disc identity to an early embryonic state[41]. By artificially over-expressing a differentiation factor, these cancer cells can be induced to lose their neoplastic state, however, it appears as if these tumour cells are trapped in an immature condition, unable to differentiate. Pulses of ecdysone are a major timer of developmental transitions in flies and their target, *let-7*, is for example required for enforcing the terminal cell cycle arrest in pupal stage wing discs[22,28]. Our findings suggest that flies have evolved tumour suppressive mechanisms by inducing *let-7*-controlled heterochronic gene networks to enforce cellular differentiation in epigenetically derailed tumours (Fig. 4i). Indeed, differentiation therapy is considered a promising approach for curing human cancers[42]. However, the strategy has been applied only in limited cases. As such, our identification of an innate tumour eviction mechanism in flies based on these principles may provide new ideas how such cancer treatments could be further improved in human patients.

## Methods

**Fly genetics.** Fly strains were maintained on standard medium. All genetic experiments were performed at 25 °C, except that the *brat*-RNAi experiments were completed at 29 °C to increase the knockdown efficiency. To test the roles of candidate genes (*let-7*, *chinmo*, etc.) in the tumorigenic growth of $ph^{505}$ mutant cells, we attempted to generate fly strains carrying double mutations including the candidate genes in combination with $ph^{505}$. However, we were unable to establish double mutant strains of $ph^{505}$ and another gene mutant. Therefore, in this study, we used transgenic fly strains expressing either RNAi or miRNA sponge to reduce the activities of the target genes.

To generate homozygous $ph^{505}$ MARCM clones, virgin females of $ph^{505}$ *FRT19A/FM7 act-GFP* were crossed with *tub-Gal80 FRT19A; ey-flp act>STOP>Gal4 UAS-GFP* males. For the control MARCM clones, virgin females of *y w FRT19A* were crossed with *tub-Gal80 FRT19A; ey-flp act>STOP>Gal4 UAS-GFP* males. Because the *ey-flp* is expressed not only in the eye-antennal discs, but also in all leg discs and the genital disc, clones were observed in these tissues as well. As a result, GFP-labelled $ph^{505}$ cells can be seen in the heads, legs, thorax, and abdomen in the adult flies. Other MARCM clones using different transgenic strains in combination with $ph^{505}$ or *FRT19A* were generated in the same way by crossing corresponding virgin females with *tub-Gal80 FRT19A; ey-flp act>STOP>Gal4 UAS-GFP* males.

**Fly strains.** The following fly strains were used in this study: (1) *Ore-R*, (2) $w^{1118}$, (3) $ph^{505}$ *FRT19A/FM7 act-GFP*, (4) *y w FRT19A*, (5) *tub-Gal80 FRT19A; ey-flp act>STOP>Gal4 UAS-GFP*, (6) *w; UAS-EcR^{DN}* (BL-9451), (7) *w;; UAS-EcR^{DN}* (BL-9450), (8) *y w; UAS-RNAi-usp* (BL-27258), (9) *w;; UAS-let-7-C* (N. Sokol), (10) *y w;; UAS-let-7* (L. Johnston), (11) *w;; UAS-miR-100* (C. Gendron), (12) *y w;; UAS-*

*miR-125* (L. Johnston), (13) *w; UAS-let-7-SP; UAS-let-7-SP/TM6B* (BL-61365), (14) *w; UAS-miR-100-SP/CyO; UAS-miR-100-SP* (BL-61391), (15) *w; UAS-miR-125-SP/ CyO; UAS-miR-125-SP* (BL-61393), (16) *y w; Pin/CyO; UAS-chinmo* (BL-50740), (17) *y w; UAS-RNAi-chinmo/TM3, Sb* (BL-33638), (18) *y w; UAS-mCherry:Atg8a; Dr/TM3 Ser* (BL-37750), (19) *w UAS-dicer2; wor-Gal4 ase-Gal80; UAS-mCD8:: GFP*, (20) *w; UAS-RNAi-brat* (VDRC-105054), (21) *w;; UAS-EcRB1* (BL-6469).

**Immunohistochemistry and antibodies.** All the tissues (larval discs, larval brains, metamorphed $ph^{505}$ cells, adult brains, and transplanted tumours) were dissected in cold PBS on ice, fixed in 2% paraformaldehyde (in 1× PBS) for 25 min at room temperature, and washed several times in PBST (1× PBS with 0.5% Triton X-100). Tissues (larval discs, larval brains, or metamorphed $ph^{505}$ cells) were incubated overnight with primary antibodies at 4 °C, followed by several washes at room temperature, incubated with secondary antibodies at 4 °C overnight. Adult brains and transplanted tumours were incubated with primary antibodies for 48 h at 4 °C, followed by several washes at room temperature, incubated with secondary antibodies for 48 h at 4 °C. After several washes, all tissues were incubated with DAPI (1:200 in PBST) at room temperature for 20 min. Tissues were then mounted in Vectashield and stored at −20 °C before analysis.

Primary antibodies used in this study were: chicken anti-green fluorescent protein (GFP) (1:1000; Abcam ab13970); mouse anti-Elav (1:30; DSHB 9F8A9); rabbit anti-cleaved DCP-1 (1:300; Asp216, Cell Signaling Technology 9578S); rabbit anti-PH3 (1:100; Millipore 06-570,); rat anti-Chinmo (1:500; from N. Sokol); mouse anti-EcRB1 (1:100; DSHB AD4.4); mouse anti-nc82 (1:50; from J. Pielage); rat anti-Dpn (1:1, from C.Y. Lee). Secondary antibodies were: Alexa 488-, Alexa 555-, Alexa 568-, and Alexa 647-conjugated anti-chicken, -rabbit, -rat, or -mouse IgG (all 1:500; Molecular Probes).

**Microscopy.** Immunofluorescent images were recorded on a confocal microscope (TCS SP5; Leica). Adult fly pictures were taken on a Leica MZ16 or a Nikon SMZ1270 microscope. Images were processed using ImageJ, Imaris, Photoshop, and Adobe Illustrator. To measure the volume of tumour clones, confocal images of eye-antennal discs from wandering third instar larvae of the corresponding genotypes were collected and processed in Imaris.

**Transplantation.** Transplantation experiments were carried out as previously described[43]. In brief, 4–6-day-old adult $w^{1118}$ females were used as hosts (*n* > 20 for each transplantation). The host flies were immobilized on an ice-cold metal plate and stuck on a piece of double-sided sticky tape, with their ventral sides up. The dissected eye discs or other tumour tissue from adult flies were cut into small pieces and each piece was transplanted into the abdomen of one host using a custom-made glass needle. All transplantation was made under a GFP microscope to ensure labelled cells were injected into the hosts. After transplantation, host flies were allowed to recover at room temperature for 1–2 h in fresh standard *Drosophila* medium before transferred to and maintained at 25 °C.

**Survivorship measurement.** To measure the survivorship of $ph^{505}$ and $ph^{505}$; *UAS-EcR^{DN}* adult flies, newly eclosed adults were collected and separated into three groups (for $ph^{505}$ flies, *n* = 20 in each group; for $ph^{505}$; *UAS-EcR^{DN}* flies, *n* = 25 in two groups and *n* = 20 for the third group). Flies were maintained at 25 °C, flipped into fresh food vials every 2 days, and the number of living flies were counted on each Monday and Friday during the course of experiment. The numbers of flies were imported into Excel and the survivorship at different times was calculated. The survival curve was produced in Excel.

**Quantitative PCR.** As previously reported[18], the expression of *let-7* is low at wandering third larval instar but high during pupal stage in response to ecdysone, we dissected eye-antennal discs together with brains from larvae (wt L3) or pupae (wt pupa) as controls. We collected metamorphed $ph^{505}$ cells from adults, transplanted tumours from hosts, $ph^{505}$; *UAS-EcR^{DN}* cells from adults, and $ph^{505}$; *UAS-let-7-SP* cells from adults. RNA was extracted from all these samples with mirVana miRNA Isolation Kit (Invitrogen), and reverse transcribed using TaqMan MicroRNA Reverse Transcription kit (Applied Biosystems) with *let-7* RT primer and 2S rRNA reference RT primer. qPCR was performed with TaqMan Small RNA Assays (Applied Biosystems) using each *let-7* and 2S rRNA TaqMan Small RNA Assay primer mix (Thermo Fisher Scientific).

**Transcriptome analyses.** Transcriptome analyses were performed by mRNA sequencing. RNAs were isolated from the metamorphed $ph^{505}$ cells, as well as RNAs from the $ph^{505}$ tumours transplanted in adult hosts after 4 times, 8 times, and 14 times weekly retransplantation. As a control, we used the wild-type discs that were transplanted into the abdomen of adult hosts and extracted RNAs 1 day after their transplantation. This step was necessary as earlier evidence showed that the transplantation process itself introduced differential expression of a number of genes[44]. RNA was extracted using an Arcturus PicoPure RNA Isolation kit (Applied Biosystems), library prepared with SMARTSeq2 NexteraXT and sequenced on an Illumina NextSeq2500 for control and tumour samples and NextSeq500 for metamorphed samples.

**Bioinformatics analyses**. Short reads were aligned to BDGP dm6 genome assembly using TopHat 2.0.12[45] with parameters "--very-sensitive" for Bowtie 2.2.3. From the aligned reads, differential expression was called using R 3.3.1 with maSigPro 1.46.0[46]. Genes were stratified into expression profile clusters with k-means clustering. Subsequent biological process gene ontologies for each cluster were found with topGO[47]. Respective *p* values were calculated with Fisher's exact test. For the hierarchical clustering (UPGMA) underlying the heat map, we used library normalized log2 counts per million reads determined by edgeR 3.16.5[48] on genes involved in the ecdysone response as listed by Flybase with scaling per sample/column.

**Experimental design**. All fly genetic experiments were repeated at least three times. Each transplantation experiment was performed at least twice independently. RNA samples for the transcriptome analysis were collected two or three times independently. RNA samples for qPCR were extracted from the mixture of 5−10 independent animals and qPCR was performed biologically twice.

Sample size was not predetermined before the experiments. Sample size in each experiment was randomized based on the number of viable flies and the amount of other materials. No data were excluded. The investigators were not blinded to group allocation during data collection and analysis.

**Data availability**. All deep-sequencing data pertaining to this study can be found on GEO GSE101455. Materials and all relevant data from this study are available from the corresponding author upon reasonable request.

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

## Acknowledgements

We are grateful to Nicholas S. Sokol, Laura A. Johnston, Christi M. Gendron, Scott D. Pletcher, Cheng-Yu Lee, and Jan Pielage for flies and antibodies. We thank Katja Eschbach in the Genomics Facility Basel and the Single Cell Facility of the D-BSSE for their excellent technical support. We also acknowledge the Bloomington *Drosophila* Stock Center and the Vienna *Drosophila* RNAi Center for fly stocks, and the Developmental Studies Hybridoma Bank for monoclonal antibodies. We thank Jorge Beira for making and sharing the *ph505; UAS-EcR^{DN}* stock, Anna Groner and Christian Beisel for comments on the manuscript and all members of the Paro group for discussions. This work was supported by the Swiss National Science Foundation and the ETH Zürich.

## Author contributions

Y.J. and M.S. designed and conducted all the experiments, under the supervision of R.P. T.B.S. and M.S. performed the bioinformatics analyses. Y.J., M.S., T.B.S. and R.P. analysed the results, created the figures, and wrote the paper.

## Additional information

**Competing interests:** The authors declare no competing interests.

