## [Peer Review File · Nature Communications]

Reviewers' comments:

Reviewer #1 (Remarks to the Author):

This is an interesting report of a possible mechanism for a steroid hormone-induced block to tumorigenesis. The authors use mutants of the *Drosophila* PRC1 complex member, polyhomeotic (*ph*) to induce neoplastic growth in clones in the eye imaginal disc during the larval stage, using the *ey-Flp* method. The *ph* mutant clones, marked with GFP, can be transplanted from L3 larvae into the abdomens of wild-type adult females where they continue to grow and become neoplastic.

The interesting observation made here is that if the larvae carrying the *ph* mutant clones are allowed to undergo metamorphosis and eclose into adult flies, the tumors are at first visible (as GFP+ cells) but over time they disappear. Transplanted *ph* mutant eye disc/brain tumors from larvae also disappear over time in wild type adults. Some of the cells show induction of markers of apoptosis and/or autophagy, suggesting that these cells are eliminated through one or both of these processes. The authors explored the role of ecdysone, which is relatively low in growing larvae but fairly high in the female abdomen due to the production in the ovaries. The essence of the findings here is that *let-7*, known to be induced by ecdysone, regulates tumorigenicity via its control of the transcription factor *chinmo*. The authors claim that the activity of *let-7* in adults turns off *chinmo* and thereby prevents the tumors from continuing to grow.

Although the work here is interesting and provocative, it seems a bit preliminary. In addition, there are a few puzzles that need to be addressed.

1. The biggest puzzle for me is that there is a substantial increase in ecdysone at the end of L3 that triggers the onset of pupariation and the beginning of metamorphosis. This is when *let-7* is first induced and active. If *let-7* regulation of *chinmo* is important for blocking the tumor growth, why are the tumors able to continue to grow at this point? Related to this, are levels of ecdysone important, e.g., is there more ecdysone and/or more *let-7* expressed in aging adults than in pupae? This needs to be addressed, otherwise the puzzle remains.
2. Where is *chinmo* expressed in the discs, and how does its expression pattern change over developmental time, e.g., in response to *let-7* expression in the pupal stage?
3. Chawala and Sokol (2016) showed that *let-7* and miR125, which is expressed in the same operon as *let-7*, regulates *chinmo* during development, whereas miR125 takes that role over in aging adults. Does miR125 have a regulatory role in the *ph505* tumors? Is this tumor eviction an aging phenomenon?
4. Abrupt is another *let-7* target that has been shown to promote tumorigenesis (Doggett et al 2015). What happens to Ab in the *ph505* tumors?
5. How effective is the *let-7* sponge at knocking down *let-7*, and how specific is it? How does it compare to the null *let-7* mutants that exist?
6. There seems to be another paper submitted that directly bears on this one, mentioned in the discussion as Torres et al. Details of data from that paper are used in an argument to support the authors' conclusions in the present paper, but obviously makes these arguments difficult to assess.

Reviewer #2 (Remarks to the Author):

In this manuscript, Jiang, Y. et al. report an intrinsic mechanism by which a molting steroid hormone (Ecdysone) suppresses tumor growth induced by loss of function of polyhemeotic genes (ph505 mutant) in *Drosophila*. Using genetic approaches, the authors showed that Ecdysone inhibited tumor growth by inducing miR Let-7-c which in turn suppresses its target gene chinmo, resulting in suppression of continuous tumor growth. This is a significant research that is well designed and well executed. The manuscript is also well written with clear presentation and description. The data is of high quality and strongly supports the overall conclusion of this study. Moreover, there are a few minor weaknesses in this study. The manuscript can be further improved if the following concerns are addressed.

1. Clinical relevance would be strengthened if the authors include data or analysis of let-7-c regulation in clinical cancer in humans.
2. In the same line, if the authors show a similar regulation of the orthologs of *Drosophila* Ecdysone/let-7-c/chinmo in human cells or human cancer cells, the clinical relevance could be further strengthened.
3. It is not clear that the tumorigenic ph505 mutant cells were injected into the abdomen of flies. The tumors were formed throughout the body including the brain. Were these tumors considered as metastatic tumors? On the other hand, these tumorigenic cells were from eye antennal disc, what is counterpart tumor in humans?

Response to Reviewers

Reviewers' comments:

Reviewer #1 (Remarks to the Author):

This is an interesting report of a possible mechanism for a steroid hormone-induced block to tumorigenesis. The authors use mutants of the Drosophila PRC1 complex member, polyhomeotic (ph) to induce neoplastic growth in clones in the eye imaginal disc during the larval stage, using the ey-Flp method. The ph mutant clones, marked with GFP, can be transplanted from L3 larvae into the abdomens of wild-type adult females where they continue to grow and become neoplastic.

The interesting observation made here is that if the larvae carrying the ph mutant clones are allowed to undergo metamorphosis and eclose into adult flies, the tumors are at first visible (as GFP+ cells) but over time they disappear. Transplanted ph mutant eye disc/brain tumors from larvae also disappear over time in wild type adults. Some of the cells show induction of markers of apoptosis and/or autophagy, suggesting that these cells are eliminated through one or both of these processes. The authors explored the role of ecdysone, which is relatively low in growing larvae but fairly high in the female abdomen due to the production in the ovaries. The essence of the findings here is that let-7, known to be induced by ecdysone, regulates tumorigenicity via its control of the transcription factor chinmo. The authors claim that the activity of let-7 in adults turns off chinmo and thereby prevents the tumors from continuing to grow.

We thank the reviewer for her/his overall positive evaluation of our manuscript and for her/his critical and helpful comments. We addressed the concerns that the reviewer raised, and incorporated these changes in our revised manuscript.

Although the work here is interesting and provocative, it seems a bit preliminary. In addition, there are a few puzzles that need to be addressed.

1. The biggest puzzle for me is that there is a substantial increase in ecdysone at the end of L3 that triggers the onset of pupariation and the beginning of metamorphosis. This is when let-7 is first induced and active. If let-7 regulation of chinmo is important for blocking the tumor growth, why are the tumors able to continue to grow at this point? Related to this, are levels of ecdysone important, e.g., is there more ecdysone and/or more let-7 expressed in aging adults than in pupae? This needs to be address, otherwise the puzzle remains.

We thank the reviewer for this comment. Unfortunately, there is a misunderstanding in regard of the activity peaks of ecdysone and let-7 during the transition of the larva (L3) to the adult stage (metamorphosis). In flies, the level of ecdysone at the end of larval stage is relatively low; it starts to increase at the beginning of metamorphosis, eventually reaching the peak level about 48 hours after the start of pupation (Fig. 4i). Afterward, the ecdysone level decreases gradually and is present only at a low basal level in adult flies. The expression of let-7 correlates with the changes of ecdysone levels. Therefore, at the end of L3, the levels of both let-7 and ecdysone are still low and not sufficient to suppress the tumour growth. In this work, we show that the transformation and suppression of tumour growth occurs during metamorphosis, when ecdysone and let-7 are at their peak heights. Indeed, at the end of the 3rd larval instar brain tumours of the *brat* mutants have reached a considerable size. After exposure to the major ecdysone peak and passage through metamorphosis brain tumours shrink resulting in almost wild-type brain phenotypes in all adult flies (see figure 4).

To clarify this concern, we now explicitly described how the expression level of ecdysone changes overtime during fly development in the revised manuscript. In addition, we also explained how the expression level of let-7 correlates with the changes of ecdysone.

2. Where is chinmo expressed in the discs, and how does its expression pattern change over developmental time, e.g., in response to let-7 expression in the pupal stage?

We carried out new immunostaining experiments to look at the expression pattern of Chinmo at the wandering larval stage and during pupal stage. At the wandering L3, we observed Chinmo was expressed in the *ph*⁵⁰⁵ mutant clones in the eye discs. At 48 hours after pupa formation, when ecdysone level was at the peak and *let-7* was highly expressed, the expression of Chinmo became undetectable, the same as in the adult metamorphosed *ph*⁵⁰⁵ cells. We have incorporated these new results in our manuscript and presented the data as a new figure in the supplementary figures (Supplementary Fig. 6).

3. *Chawala and Sokol (2016) showed that let-7 and miR125, which is expressed in the same operon as let-7, regulates chinmo during development, whereas miR125 takes that role over in aging adults. Does miR125 have a regulatory role in the ph505 tumors? Is this tumor eviction an aging phenomenon?*

We thank the reviewer for this interesting question. Indeed, the *let-7-C* locus in *Drosophila* also encodes, besides *let-7*, another two miRNAs, *mir-100* and *mir-125* (as in the human chromosome 11 cluster). However, the expression patterns, transcriptional regulation, and biological functions of these two miRNAs are largely unknown. In order to assess a potential involvement in the tumour reprogramming we carried out new genetic experiments and used transgenic sponge to reduce their activities in *ph*⁵⁰⁵ cells (*ph*⁵⁰⁵; *UAS-mir100-SP* and *ph*⁵⁰⁵; *UAS-mir125-SP*). We found that, both *ph*⁵⁰⁵; *UAS-mir100-SP* and *ph*⁵⁰⁵; *UAS-mir125-SP* cells did not continue to grow in the adult flies and hence remained metamorphosed. This is a different outcome than the *let-7-SP* experiment, suggesting that *mir-100* and *mir-125* are most likely not required to suppress tumourigenic growth of *ph*⁵⁰⁵ cells. We described these results in the text of the revised manuscript. As these experiments disclosed a negative result, we decided not to show the pictures.

4. *Abrupt is another let-7 target that has been shown to promote tumorigenesis (Doggett et al 2015). What happens to Ab in the ph505 tumors?*

We had already analyzed the expression of Abrupt by immunostaining with Ab antibodies, and did not detect the expression of Ab in all the different samples, including the transplanted tumours, the eye-antennal discs, as well as the metamorphosed *ph*⁵⁰⁵ cells. Since these results were negative, we did not include them in our first manuscript. We thank the reviewer for raising this question. We now point out the lack of Abrupt expression in our revised manuscript.

5. *How effective is the let-7 sponge at knocking down let-7, and how specific is it? How does it compare to the null let-7 mutants that exist?*

We thank the reviewer for raising this concern. Indeed, we tried to use mutants of *let-7*, as well as other genes (*chinmo*, *usp*, etc.) in our study. However, we were unable to generate double mutant fly strains combining both the *ph*⁵⁰⁵ mutant with other gene mutants. Therefore, throughout this study, we had to use transgenic strains expressing either RNAi or miRNA sponge to knock down the target genes and tested their roles in the development and transformation of *ph*⁵⁰⁵ tumours. To clarify this concern, we added text in the Method section, to explain why we did not use the gene mutants in our study. To test the efficiency of the *let-7* sponge, we did quantitative PCR measurement using the Taqman miRNA assay and showed that the expression level of *let-7* was substantially reduced (Fig. 3a). To further confirm this result, we also did a biological replicate measurement using newly collected tissues and cells. We now incorporated our new data in the revised manuscript as Figure 3a.

6. *There seems to be another paper submitted that directly bears on this one, mentioned in the discussion as Torres et al. Details of data from that paper are used in an argument to support the authors' conclusions in the present paper, but obviously makes these arguments difficult to assess.*

We thank the reviewer for the interest in the other related manuscript from our lab. The other manuscript is currently under review at another journal. In essence, the work identifies a single transcription factor, Knirps, as a new oncogene capable of activating the JAK/STAT signaling pathway in *ph* tumour cells. This prevents the tumour cells to undergo

differentiation. By artificially overexpressing a differentiation gene (atonal) the tumourigenic capacity is reduced.

In this current work, we now show that *Drosophila* has apparently developed an innate mechanism to force differentiation onto tumour cells during metamorphosis (by the ecdysone/let-7 pulse during metamorphosis).

The summary of the manuscript under review is pasted below. We would be happy to provide the editor with a copy of the manuscript, if requested.

A switch in transcription and cell fate governs the onset of an epigenetically-deregulated tumor in Drosophila

Torres et al.

Summary

Tumor initiation is often linked to a loss of cellular identity. Transcriptional programs determining cellular identity are preserved by epigenetically-acting chromatin factors. Although such regulators are among the most frequently mutated genes in cancer, it is not well understood how an abnormal epigenetic condition contributes to tumor onset. In this work, we investigated the gene signature of tumors caused by disruption of the *Drosophila* epigenetic regulator, *polyhomeotic* (*ph*). In larval tissue *ph* mutant cells show a shift towards an embryonic-like signature. Using loss- and gain-of-function experiments we uncovered the embryonic transcription factor *knirps* (*kni*) as a new oncogene. The oncogenic potential of *kni* lies in its ability to activate JAK/STAT signaling and block differentiation. Conversely, tumor growth in *ph* mutant cells can be substantially reduced by overexpressing a differentiation factor. This demonstrates that epigenetically derailed tumor conditions can be suppressed by induction of differentiation.

Reviewer #2 (Remarks to the Author):

In this manuscript, Jiang, Y. et al. report an intrinsic mechanism by which a molting steroid hormone (Ecdysone) suppresses tumor growth induced by loss of function of polyhemeotic genes (ph505 mutant) in Drosophila. Using genetic approaches, the authors showed that Ecdysone inhibited tumor growth by inducing miR Let-7-c which in turn suppresses its target gene chinmo, resulting in suppression of continuous tumor growth. This is a significant research that is well designed and well executed. The manuscript is also well written with clear presentation and description. The data is of high quality and strongly supports the overall conclusion of this study. Moreover, there are a few minor weaknesses in this study. The manuscript can be further improved if the following concerns are addressed.

We are very grateful to the reviewer for her/his positive assessment of our manuscript and the constructive advice. Indeed, adding more information to connect our finding from Drosophila with more clinically relevant human cases would certainly be very valuable for a general readership. The comments addressed by the reviewer are discussed below.

1. Clinical relevance would be strengthened if the authors include data or analysis of let-7-c regulation in clinical cancer in humans.

2. In the same line, if the authors show a similar regulation of the orthologs of Drosophila Ecdysone/let-7-c/chinmo in human cells or human cancer cells, the clinical relevance could be further strengthened.

Because both points deal with the same subject, we wish to address them together. *let-7* has been shown to be down-regulated in many human cancers. Conversely, in vitro experiments have demonstrated that over-expression of *let-7* in some cancer cell lines can suppress cell proliferation and tumour growth. The human genome encodes a *let-7* family that consists of more than 10 different members, but the transcriptional regulation, spatial and temporal expression, and their tissue-specific and/or redundant functions of any human *let-7* member remain unknown. Since ecdysone is an insect-specific molting hormone, and *chinmo* does not have a conserved human ortholog, a direct evolutionary comparison is not possible. However, the organization of the *let-7* complex (including mir-100 and mir-125) is highly conserved between flies and mammals, hence, we speculate that a similar network of heterochronic genes might act in humans. For example, the activation of the *let-7* complex might utilize comparable sets of inductive signals (hormones, small molecules, etc.). However, identifying and dissecting such a mechanism in a mammalian organism poses a substantial challenge and would be beyond the scope of this work. In the revised manuscript, we now extended the discussions in the last paragraph to discuss the potential application of our work in human cancer studies.

3. It is not clear that the tumorigenic ph505 mutant cells were injected into the abdomen of flies. The tumors were formed throughout the body including the brain. Were these tumors considered as metastatic tumors? On the other hand, these tumorigenic cells were from eye antennal disc, what is counterpart tumor in humans?

Currently, there is no indication that *ph*⁵⁰⁵ cells do give rise to metastatic tumours. The observed GFP-labeled cells throughout the body were clones generated in the leg discs and the genital discs, due to the expression of *ey-flp* in those tissues. To clarify this point, we now added texts in the 1st paragraph, as well as in the Method section in the revised manuscript to explain how the GFP-labeled cells were present all over the body.

While the imaginal discs of *Drosophila* do not have a direct counterpart tissue/organ in humans, the discs consist of a single layer of epithelial cells, which share many common features as mammalian epithelia. Therefore, they have been used as a powerful model during the past decades to investigate mechanisms underlying the formation and progression of several types of tumour, including Ras-, PcG-, or Hippo pathway-driven tumours (Gonzalez, 2013). We now added a new paragraph at the end of the revised manuscript, to describe the

use of imaginal discs as a model to study tumour and discuss the potential relevance of our work in studying the PcG-related tumours in humans.

Gonzalez, C. (2013). *Drosophila melanogaster*: a model and a tool to investigate malignancy and identify new therapeutics. *Nat. Rev. Cancer* 13, 172–183.

Reviewers' comments:

Reviewer #1 (Remarks to the Author):

The authors have responded most of the reviewers questions well. Two remaining issues that should be resolved:

1. The authors refer to Fig. 4i regarding the increase in ecdysone at the larval-pupal transition. This drawing is a simplified schema, and needs a reference as to where the data it is illustrating comes from (for ecdysone and let-7). I presume the ecdysone is meant to be 20-HE titer in the whole-animal? Presumably the chinmo and let-7 expression reflects mRNA or protein in eye discs. This needs to be made more clear in the figure legend.

2. The authors should demonstrate that the sponges for miR-100 and miR-125 are also functional, as they did for let-7.

Reviewer #2 (Remarks to the Author):

In this revised manuscript, the authors addressed most comments by two reviewers with new data and edited text. This is an improved study. However, there are still two following concerns that need to be addressed.

1. Clinical relevance of let-7-c expression in clinical cancer in humans was not addressed. The replies by the authors to this comment was not sufficient. The authors need to perform Kaplan Meir analysis for correlation of let-7-c expression with prognosis of patients with brain tumours that the authors studied in this manuscript in *Drosophila*.

2. The authors described that RNAi knockdown of either miR-100 or miR125-1 did not showed any phenotypes in brain tumour formation However, since Let-7-c, miR-100 and miR125-1 are encoded in the same chromosome locus and highly conserved from insects to humans, and miR-125-1 regulates tumorigenicity in humans by targeting Wnt signaling (Emmrich, S. et al. 2014, *Genes & Dev.*; Huang, T. et al, 2016, *Nat Comms*), these negative data should be included as supplementary figures and a discussion for these observations should be also added.

Response to Reviewers

Reviewers' comments:

Reviewer #1 (Remarks to the Author):

The authors have responded most of the reviewers questions well. Two remaining issues that should be resolved:

1. The authors refer to Fig. 4i regarding the increase in ecdysone at the larval-pupal transition. This drawing is a simplified schema, and needs a reference as to where the data it is illustrating comes from (for ecdysone and let-7). I presume the ecdysone is meant to be 20-HE titer in the whole-animal? Presumably the chinmo and let-7 expression reflects mRNA or protein in eye discs. This needs to be made more clear in the figure legend.

We thank the reviewer for this suggestion. To make our schematic drawing (Fig. 4i) clearer, we now include the following texts in the figure legend: "In the drawing, ecdysone refers to 20-HE titers during pre-pupal and pupal stages (ref. 11 & 12); *chinmo* and *let-7* refer to the mRNA expression in the eye-antenna discs."

2. The authors should demonstrate that the sponges for miR-100 and miR-125 are also functional, as they did for let-7.

We thank the reviewer for pointing out this issue. To further characterize the potential role of mir-100 and mir-125 in the transformation of ph^{505} tumours, we carried out extensive additional analyses by transgenic miRNA sponges, as well as over-expression of each individual miRNA. The results are presented in the revised manuscript as Supplementary Figure 6, and we add text to describe the details of our analyses. Our results showed that:

For mir-100: 1) ph^{505} ; mir-100-SP cells did not continue to grow and was eliminated by apoptosis in the adults; 2) quantitative PCR showed that the expression of mir-100 did not decrease in the mir-100-SP cells; 3) over-expression of mir-100 did not suppress the growth of ph^{505} tumours in the eye-antennal discs.

For mir-125: 1) ph^{505} ; mir-125-SP cells did not continue to grow in the adult flies too; 2) however, we observed an unexpected phenotype that ph^{505} ; mir-125-SP cells did not die and lacked expression of apoptosis cell marker; 3) adult ph^{505} ; mir-125-SP cells also are not tumorigenic, as they did not give rise to tumours after transplantation into host flies; 4) quantitative PCR measurement showed that the expression level of mir-125 did not reduce in the ph^{505} ; mir-125-SP cells; 5) conversely, we found that mir-125-SP resulted in an increase in the expression of the entire let-7-C primary transcript, indicating a possible feed-back and cross regulation mechanism among these three miRNAs; 6) over-expression of mir-125 only slightly reduced the growth of ph^{505} cells in the eye-antennal discs, and after transplantation, ph^{505} ; UAS-mir-125 discs still gave rise to the formation of tumours in a substantial subset of hosts.

As the quantitative RT-PCR analyses did not show decreased level of either mir-100 or mir-125 by the transgenic sponges, superficially it appeared that both sponges were not functional. When we measured the primary transcripts of the whole let-7-C locus, we found that let-7-SP did not change the expression of the let-7-C. Conversely, mir-125-SP increased the levels of the entire locus, which could result in

further complex consequences in the cell and lead to the observed phenotype of ph^{505} ; mir-125-SP cells in the adult flies, however. To further evaluate the roles of mir-100 and mir-125, we next carried out over-expression experiments. Over-expression of mir-100 did not suppress the growth of ph^{505} cells in the eye-antennal discs, while overexpression of mir-125 moderately reduced the tumourigenicity.

In summary, we believe that let-7 is the key regulator in the let-7-C locus that is both necessary and sufficient to control the transformation of ph^{505} cells during metamorphosis, while the other two miRNAs might have minor contributions. However, due to the cross regulations among these three miRNAs, and the limitation of available genetic tools, we were unable to further dissect the function of the individual miRNA without affecting the others.

Reviewer #2 (Remarks to the Author):

In this revised manuscript, the authors addressed most comments by two reviewers with new data and edited text. This is an improved study. However, there are still two following concerns that need to be addressed.

1. Clinical relevance of let-7-c expression in clinical cancer in humans was not addressed. The replies by the authors to this comment was not sufficient. The authors need to perform Kaplan Meir analysis for correlation of let-7-c expression with prognosis of patients with brain tumours that the authors studied in this manuscript in Drosophila.

We agree with the reviewer that it would be more interesting to translate our findings in *Drosophila* to the clinical human cancer patients. However, clinical studies with human patient samples are much more complicated and far beyond the scope of our manuscript. As we've discussed in our manuscript, the human genome has more than ten members of the let-7 family, whose transcriptional regulation, spatial and temporal expression, as well as functions are mostly unknown. In addition, different let-7 members may play different and/or redundant roles in different types of human cancers.

We do also find changes in Wnt/wg expression when comparing *ph* tumor cells with metamorphosed cells, as pointed out by the reviewer (Emmrich, S. et al. 2014, *Genes & Dev.*; Huang, T. et al, 2016, *Nat. Comms*). However, we believe a direct comparison and discussion at this molecular level to be premature given the highly diverse biological materials considered. Similarly, at this stage of the work a direct correlation of fly brain tumour physiology with a human brain tumour condition would be arbitrary and in our opinion not informative.

Our work shows that the network of heterochronic genes and in particular the conserved let-7 component transforms a tumorous state into a differentiated (metamorphosed) non-tumorous state. In flies this is achieved through the "bottleneck" metamorphosis where a natural mechanism appears to force cancer cells into differentiation. In Torres et al. (2018) we demonstrated that by artificially overexpressing a differentiation marker the tumourigenicity of ph^{505} cell can be suppressed. Indeed, "differentiation therapy" is considered a promising approach for curing human cancers (de Thé (2018)). However, the strategy has been applied only in limited cases. Our findings offer new and additional routes how such therapies could eventually be implemented. Hence, we believe that our work will be interesting to the broad readership of *Nature Communications*.

2. *The authors described that RNAi knockdown of either miR-100 or miR125-1 did not show any phenotypes in brain tumour formation. However, since Let-7-c, miR-100 and miR125-1 are encoded in the same chromosome locus and highly conserved from insects to humans, and miR-125-1 regulates tumorigenicity in humans by targeting Wnt signaling (Emmrich, S. et al. 2014, Genes & Dev.; Huang, T. et al, 2016, Nat Comms), these negative data should be included as supplementary figures and a discussion for these observations should be also added.*

We thank the reviewer for pointing out this issue. We carried out extensive analyses to evaluate the potential role of miR-100 and miR-125 (for details see also response to reviewer 1). Our results are now presented in the revised manuscript as Supplementary Figure 6, and we add additional texts to describe the details of our analyses

References:

Torres, J., Monti, R., Moore, A.L., Seimiya, M., Jiang, Y., Beerenwinkel, N., Beisel, C., Beira, J.V., and Paro, R. (2018). A switch in transcription and cell fate governs the onset of an epigenetically-deregulated tumor in *Drosophila*. *Elife* 7, 32697.

de Thé, H. (2018). Differentiation therapy revisited. *Nat Rev Cancer* 18, 117–127.

Reviewers' comments:

Reviewer #1 (Remarks to the Author):

The authors have made significant improvements to their ms. Although it is very close to being ready for publication, I still have concerns about the analysis of miR functional roles, and some further clarification is needed with some of the text.

1. One issue is that it is not clear to me why the authors look at the expression of the miR in question to determine the functional efficiency of its sponge, rather than a known target of the miR. Since sponges competitively sequester specific miRNAs to prevent miRNA/mRNA interaction, they should not reduce the level of the miR itself. Rather, targets of the miR should be up-regulated when the sponge works well. If a target isn't known (e.g., for miR-100 and miR-125), one can look for a phenotype that is known from null alleles of that miR. miR sponges are prone to off target effects (also true for miR over-expression) so it seems important to make sure they work appropriately. And although the let-7, miR-100 and miR-125 sponges used in this work were obtained from the BDSC, to my knowledge these specific sponges have not been tested for functionality. The authors want to make the case that let-7, but not miR100 nor miR-125, is causal in the extinction of the ph505 tumors in adults. Since the data presented on mir-100 and miR-125 are not straightforward (and indeed are somewhat complicated by the different phenotypes they observed for each), they don't yield easy conclusions. Some of this might be cleared up by verifying that the sponges do (or don't) work, and if the latter, the data on those miRs might not be worth including. Nonetheless, it is important to definitively show the requirement for let-7, and also that the let-7 sponge works accurately on known targets. For example, does the let-7 sponge specifically cause upregulation of Chinmo or Abrupt under conditions where either is normally expressed?

2. At several points in the ms. there are statements made that are ambiguous or are missing back-up data.

- Line 123: "not in the transplanted normal tumours (Fig. 3a)". What are "normal" tumours? In fig. 3a the tumours are called "ph505 trpl tumor".

- Line 133-134: "average tumour volume was reduced to 21% (Fig. 3d), in a cell death-independent manner (Supplementary Fig. 5)." Where is the data showing the death independence?

- Line 166-167: "Overexpression of mir-125 slightly suppressed tumour growth in the eye-antennal discs (Supplementary Fig. 6I)."

What should the image in Supp Fig 6I be compared to? Is there quantification that shows a reduction? They look similar in size to the image in Fig. 1a.

- In Fig. 2c: it would help if the different lines graphed were defined in the legend. Are they independent experiments of the same genotype? Also, in Fig. 2e, the shading implies error bars but no explanation is given. What do they represent and how were they calculated?

Reviewer #2 (Remarks to the Author):

The authors addressed my comments. However, it is imperative to establish clinical relevance of let-7-c expression with prognostic values of patients with gliomas. Let-7-c was actually found expressed at relative higher levels in a GBM subtype compared to other subtypes GBM tumors. It will be highly informative to perform analysis of clinical significance of let-7-c using TCGA RNA seq data of LGG-GBM that is available online.

Response to Reviewers (NCOMMS-17-19962B)

Reviewers' comments in italics/response in blue

Reviewer #1:

The authors have made significant improvements to their ms. Although it is very close to being ready for publication, I still have concerns about the analysis of miR functional roles, and some further clarification is needed with some of the text.

1. One issue is that it is not clear to me why the authors look at the expression of the miR in question to determine the functional efficiency of its sponge, rather than a known target of the miR. Since sponges competitively sequester specific miRNAs to prevent miRNA/mRNA interaction, they should not reduce the level of the miR itself. Rather, targets of the miR should be up-regulated when the sponge works well. If a target isn't known (e.g., for miR-100 and miR-125), one can look for a phenotype that is known from null alleles of that miR. miR sponges are prone to off target effects (also true for miR over-expression) so it seems important to make sure they work appropriately. And although the let-7, miR-100 and miR-125 sponges used in this work were obtained from the BDSC, to my knowledge these specific sponges have not been tested for functionality. The authors want to make the case that let-7, but not miR100 nor miR-125, is causal in the extinction of the ph505 tumors in adults. Since the data presented on mir-100 and miR-125 are not straightforward (and indeed are somewhat complicated by the different phenotypes they observed for each), they don't yield easy conclusions. Some of this might be cleared up by verifying that the sponges do (or don't) work, and if the latter, the data on those miRs might not be worth including. Nonetheless, it is important to definitively show the requirement for let-7, and also that the let-7 sponge works accurately on known targets. For example, does the let-7 sponge specifically cause upregulation of Chinmo or Abrupt under conditions where either is normally expressed?

We thank the reviewer for the valuable suggestion to clarify the functional efficiency of the three individual miRNA sponges. We agree with the reviewer that it is more appropriate to check the expression of the miRNA targets in the different miRNA sponge-expressing cells rather than the levels of the miRNAs. Therefore we carried out new quantitative PCR analyses to check the expression of some miRNA targets. As previously reported, *chinmo* is a known target of *let-7* as well as *miR-125* in the fly nervous system. Our new qPCR analyses showed that, the expression of *chinmo* is indeed increased in the *ph⁵⁰⁵; UAS-let-7-SP* cells, confirming that *let-7-SP* is functional. We presented this result as a new figure in the manuscript (Supple. Fig. 6h). On the other hand, *chinmo* was only weakly up-regulated in the *ph⁵⁰⁵; UAS-miR-125-SP* cells (Supple. Fig. 6i), most likely due to the repression of *let-7* in these cells. As the reviewer pointed out, there are no other well-characterized targets for *miR-125* and *miR-100*. Therefore, we performed computational analyses and chose predicted target genes for qPCR measurements. Our analyses showed that, the expression of two *miR-125* predicted targets were up-regulated in the *ph⁵⁰⁵; UAS-miR-125-SP* cells (Supple. Fig. 6i), but the expression of the *miR-100* targets did not

increase in *ph*⁵⁰⁵; *UAS-miR-100-SP* cells. These results indicate that miR-125-SP is working, but miR-100-SP is apparently not.

Taken together, based on both the sponge analyses, as well as the over-expression experiments, the results suggest that *let-7* is the key regulator in the *let-7-C* locus that is both necessary and sufficient for the suppressing the tumourigenic character of *ph*⁵⁰⁵ cells during metamorphosis. While we did not identify a significant involvement of *miR-100*, *miR-125* may perform a certain contribution in the elimination of the metamorphed cells, as we observed the phenotype in the sponge experiments. As previously reported, we also observed complicated cross regulations among the three individual miRNAs during our analyses. Due to the limitation of available genetic tools and methods, at this moment, we realize it is impossible to further dissect the contribution of each individual miRNA without affecting the other two in the *let-7-C* locus.

2. At several points in the ms. there are statements made that are ambiguous or are missing back-up data.

- Line 123: “not in the transplanted normal tumours (Fig. 3a)”. What are “normal” tumours? In fig. 3a the tumours are called “*ph*⁵⁰⁵ trpl tumor”.

We thank the reviewer and changed the text in Line 123 to “*ph*⁵⁰⁵ transplanted tumours”.

- Line 133-134: “average tumour volume was reduced to 21% (Fig. 3d), in a cell death-independent manner (Supplementary Fig. 5).” Where is the data showing the death independence?

We thank the reviewer for this comment. We observed a significant reduction in the tumour volume when over-expressing *let-7*. To clarify that such a reduction was not due to cell death, we performed new immunostaining experiments using antibodies labelling apoptotic cells. We found, when *let-7* was over-expressed in the eye-antennal discs, the majority of the cells within a GFP-labeled clone did not show the expression of the cell death marker cDCP-1, indicating over-expression of *let-7* did not induce cell death. This result is now presented in the revised manuscript as a new figure (Suppl. Figure 5d).

- Line 166-167: “Overexpression of *mir-125* slightly suppressed tumour growth in the eye-antennal discs (Supplementary Fig. 6l).” What should the image in Supp Fig 6l be compared to? Is there quantification that shows a reduction? They look similar in size to the image in Fig. 1a.

We did quantification of the tumour volume in the eye-antennal discs of different genotypes and show the results in the revised manuscript as a new figure (Suppl. Figure 5a). As the quantification shows, either over-expression of the whole *let-7-C* or *let-7* alone was sufficient to significantly reduce the neoplastic growth. Although over-expression of *miR-125* or *miR-100* could partially reduced the tumour volume in the eye-antennal discs, these cells still gave rise to tumour growth at a high

frequency in the transplantation experiments, suggesting over-expression of *miR-125* or *miR-100* alone was not sufficient to alter the tumorigenicity of *ph⁵⁰⁵* cells. We agree with the reviewer that the confocal images showing the tumour clones in our previous version of the manuscript are less informative, as they looked similar in size, therefore we decide not to show them in the current manuscript.

• *In Fig. 2c: it would help if the different lines graphed were defined in the legend. Are they independent experiments of the same genotype? Also, in Fig. 2e, the shading implies error bars but no explanation is given. What do they represent and how were they calculated?*

We thank the reviewer and added texts in the figure legend to explain what the shading implies.

Reviewer #2 (Remarks to the Author):

The authors addressed my comments. However, it is imperative to establish clinical relevance of let-7-c expression with prognostic values of patients with gliomas. Let-7-c was actually found expressed at relative higher levels in a GBM subtype compared to other subtypes GBM tumors. It will be highly informative to perform analysis of clinical significance of let-7-c using TCGA RNA seq data of LGG-GBM that is available online.

A meaningful direct comparison between *Drosophila* and human would require detailed knowledge of the tumor formation process and cellular origin. The insight into both conditions is currently still incomplete. As such, even when correlating strongly, the information gained by this analysis may not actually be based on an underlying similar biological process. Nonetheless, we have compared differential miRNA expression to overall survivability of patients found within the TCGA Lower Grade Glioma (LGG) datasets [1]. The human genomic locus most closely related to *Drosophila* let-7-Complex can be found on chromosome 11 q24.1, encoding hsa-miR-100, hsa-let-7a-2 and hsa-miR-125b-1 corresponding to dme-miR-100, dme-let-7 and dme-miR-125, respectively. Patients were separated into classes of lowest 50% of miRNA expression and highest 20%.

A. TCGA Lower Grade Glioma Data

Higher expression of hsa-let-7a-2 and hsa-miR-100 significantly correlates with a decrease in survival probability when compared to tumors with lower expression. Since other groups have reported an inverse relation [2], we decided to perform an analogous analysis on data generated as part of the Chinese Glioma Genome Atlas

(CGGA) [3, among others]. Notably, the CGGA data is not focused on LGG and contains Gliomas of all grades.

B. CGGA Glioma Data

Data from both TCGA and CGGA agree on higher has-miR-100 expression correlating significantly at the 0.05 level with decrease survival probability and for both data sets no statement can be made on has-miR-125b and b-1 in TCGA data. However, for CGGA data, high hsa-let-7a strongly corresponds to higher survival probability.

C. CGGA data excluding grade IV Gliomas

These disparaging results are not caused by the inclusion of grade IV Gliomas in the CGGA data, but more likely a consequence of the intrinsic complexity of Gliomas and hsa-let-7 transcriptional and post-transcriptional regulation, which is not within the scope of the manuscript. As such, we cannot support a comparison of let-7 biological functionality between *Drosophila* and human brain tumors.

[1] TCGA, *Comprehensive and Integrative Genomic Characterization of Diffuse Lower Grade Gliomas*. NEJM. **2015** 372: 2481-2498
<https://www.nejm.org/doi/full/10.1056/NEJMoa1402121>

[2] Wang X.R., Luo H., Li H.L., Cao L., Wang X.F., Yan W., Wang Y.Y., Zhang J.X., Jiang T., Kang C.S., Liu N., You Y.P.; Chinese Glioma Cooperative Group (CGCG), *Overexpressed let-7a inhibits glioma cell malignancy by directly targeting K-ras, independently of PTEN*. Neuro Oncol. **2013** 15:1491-1501
<https://academic.oup.com/neuro-oncology/article/15/11/1491/1051308>

[3] Li Y., Xu J., Chen H., Bai J., Li S., Zhao Z., Shao T., Jiang T., Ren H., Kang C., and Li X., *Comprehensive analysis of the functional microRNA-mRNA regulatory network identifies miRNA signatures associated with glioma malignant progression*. Nucleic Acids Res, **2013**. 41(22): e203
<http://www.cgga.org.cn>

REVIEWERS' COMMENTS:

Reviewer #2 (Remarks to the Author):

In response to my last comment, the authors made efforts to perform analysis of clinical relevance of let-7-c expression with prognostic values of patients with gliomas using TCGA and CGTA datasets. Although the resulted data are inclusive, at least the authors showed strong correlation between high levels of miR-100 and poor survival in gliomas. Moreover, expression of human hsa-miR-125b did not show correlation with poor survival in both datasets. In fact, a previous study also failed to show the correlation of expression of human miR-125b with patient survival when analyzing TCGA data (Huang T et al, NCOMMS. 2016, 7:12885). Thus, analysis of miR expression only may or may not yield positive clinical relevance due to the heterogeneity and complexity of gliomas. I think the response by the authors is sufficient to my concern.

Response to Reviewers (NCOMMS-17-19962C)

Reviewers' comments in italics/response in blue

Reviewer #2 (Remarks to the Author):

In response to my last comment, the authors made efforts to perform analysis of clinical relevance of let-7-c expression with prognostic values of patients with gliomas using TCGA and CGTA datasets. Although the resulted data are inclusive, at least the authors showed strong correlation between high levels of miR-100 and poor survival in gliomas. Moreover, expression of human hsa-miR-125b did not show correlation with poor survival in both datasets. In fact, a previous study also failed to show the correlation of expression of human miR-125b with patient survival when analyzing TCGA data (Huang T et al, NCOMMS. 2016, 7:12885). Thus, analysis of miR expression only may or may not yield positive clinical relevance due to the heterogeneity and complexity of gliomas. I think the response by the authors is sufficient to my concern.

We thank the reviewer again for her/his positive comments on our analysis with the clinical data.